# Genome-wide discovery for diabetes-dependent triglycerides-associated loci

**Margaret Sunitha Selvaraj**[1,2,3], **Kaavya Paruchuri**[1,2], **Sara Haidermota**[1,2], **Rachel Bernardo**[1,2], **Stephen S. Rich**[4], **Gina M. Peloso**[5‡]*, **Pradeep Natarajan**[1,2,3‡]*

1 Cardiovascular Research Center, Massachusetts General Hospital, Boston, MA, United States of America, 2 Program in Medical and Population Genetics, Broad Institute of Harvard and MIT, Cambridge, MA, United States of America, 3 Department of Medicine, Harvard Medical School, Boston, MA, United States of America, 4 Center for Public Health Genomics, University of Virginia, Charlottesville, VA, United States of America, 5 Department of Biostatistics, Boston University School of Public Health, Boston, MA, United States of America

‡ GMP and PN are jointly supervised the work.
* pnatarajan@mgh.harvard.edu (PN); gpeloso@bu.edu (GMP)

**Data Availability Statement:** All relevant data are within the paper and its Supporting Information files.

## Abstract

### Purpose

We aimed to discover loci associated with triglyceride (TG) levels in the context of type 2 diabetes (T2D). We conducted a genome-wide association study (GWAS) in 424,120 genotyped participants of the UK Biobank (UKB) with T2D status and TG levels.

### Methods

We stratified the cohort based on T2D status and conducted association analyses of TG levels for genetic variants with minor allele count (MAC) at least 20 in each stratum. Effect differences of genetic variants by T2D status were determined by Cochran's Q-test and we validated the significantly associated variants in the Mass General Brigham Biobank (MGBB).

### Results

Among 21,176 T2D and 402,944 non-T2D samples from UKB, stratified GWAS identified 19 and 315 genomic risk loci significantly associated with TG levels, respectively. Only chr6p21.32 exhibited genome-wide significant heterogeneity ($I^2$ = 98.4%; $p_{heterogeneity}$ = 2.1x$10^{-15}$), with log(TG) effect estimates of -0.066 (95%CI: -0.082, -0.050) and 0.002 (95% CI: -0.002, 0.006) for T2D and non-T2D, respectively. The lead variant rs9274619:A (allele frequency 0.095) is located 2Kb upstream of the *HLA-DQB1* gene, between *HLA-DQB1* and *HLA-DQA2* genes. We replicated this finding among 25,137 participants (6,951 T2D cases) of MGBB ($p_{heterogeneity}$ = 9.5x$10^{-3}$). Phenome-wide interaction association analyses showed that the lead variant was strongly associated with a concomitant diagnosis of type 1 diabetes (T1D) as well as diabetes-associated complications.

**Funding:** P.N. is supported by grants from the National Institutes of Health (R01HL142711, R01HL148050, R01HL151283, R01HL127564, R01HL148565, R01HL151152, R01DK125782), Fondation Leducq (TNE-18CVD04), and Massachusetts General Hospital (Fireman Chair). GMP is supported by NIH grants R01HL127564 and R01HL142711. The funders had no role in study design, data collection and analysis, decision to publish, or preparation of the manuscript

**Competing interests:** P.N. reports grants from Amgen, Apple, AstraZeneca, Boston Scientific, and Novartis, personal fees from Apple, AstraZeneca, Blackstone Life Sciences, Foresite Labs, Genentech / Roche, Novartis, and TenSixteen Bio, equity in geneXwell, and TenSixteen Bio, co-founder of TenSixteen Bio, and spousal employment at Vertex, all unrelated to the present work. This does not alter our adherence to PLOS ONE policies on sharing data and materials.

## Conclusion

In conclusion, we identified an intergenic variant near *HLA-DQB1/DQA2* significantly associates with decreased triglycerides only among those with T2D and highlights an immune overlap with T1D.

## Introduction

Diabetes, largely due to type 2 diabetes (T2D), was estimated to afflict 9.3% of population in 2019 and projected to increase to 10.9% by 2045 [1]. Despite ongoing scientific advances [2], T2D remains a leading cause of morbidity and mortality in the US and increasingly worldwide [3, 4]. Novel approaches to discover the factors influencing T2D-related metabolic alterations may yield new insights toward the prevention of T2D-related complications.

Plasma lipid, particularly triglycerides (TG), alterations represent early metabolic changes linked to insulin resistance. Hypertriglyceridemia is often observed among individuals at risk for T2D and is more severe among individuals with poorly controlled T2D [5]. Enhanced hepatic secretion of TG rich lipoproteins due to insulin resistance and delayed clearance involving lipoprotein lipase-mediated lipolysis may further exacerbate hypertriglyceridemia [6]. Hypertriglyceridemia is an independent predictor of cardiovascular disease in T2D [7, 8], as well as a predictor of T2D itself [9]. Characterizing the genetic determinants of TG concentrations specific to those with T2D may yield new insights into diabetes pathogenesis and complications.

Here, we tested the hypothesis that there are genetic variants associated with TG levels specific to T2D using GWAS and heterogeneity analysis in 424,120 participants of the UKB. Further, we assessed the role of the identified lead variant for multiple diabetes-related phenotypes.

## Results

### Baseline characteristics

The overall study schematic is depicted in **S1 Fig** Among 424,120 and 25,137 included samples, 21,176 (5.0%) and 6,951 (27.7%) of samples had T2D in UKB and MGBB, respectively. Overall UKB was composed of participants with a mean age (standard deviation [SD]) of 56.6 (8.1) years, 195,966 (46.2%) male, and 356,023 (83.9%) White British self-reported race. MGBB participants were mean 62.1 (16.2) years, 11,579 (46.1%) male, and 21,172 (84.2%) White British self-reported race. As expected, individuals with T2D versus non-T2D had greater median TG concentrations in both cohorts (**Table 1**).

### T2D-stratified GWAS of TG identified an associated locus on chromosome 6

We performed GWAS on normalized natural log TG stratified by T2D status in the discovery cohort. Among the 402,944 non-T2D samples, 315 significant loci were identified. Among the 21,176 T2D samples, 19 significant loci were identified (**S2 Fig** and **S1 Table**). We then assessed for differential TG effects for 67M variants by T2D status using Cochran's Q-test for heterogeneity. We identified 478 variants which were genome-wide significant, all at chr6p21.32 (lead variant: rs9274619:G>A; $I^2$ = 98.4%; $p_{heterogeneity}$ = 2x10$^{-15}$) (**Fig 1**). The most significantly heterogenous variant was an intergenic variant near the *HLA-DQB1/DQA2* genes

**Table 1. Baseline characteristics for discovery and replication cohorts.** Distribution of samples across the T2D strata in discovery (UKB) and replication (MGBB) cohorts are provided. Number of samples by gender, ancestry and lipid lowering medications are documented. Lipid measurements for the four main lipid class is tabulated based on T2D strata.

| Cohorts | UK Biobank | | Mass General Brigham Biobank | |
|---|---|---|---|---|
| Strata | T2D | non-T2D | T2D | non-T2D |
| Number of samples (%) | 21176(4.99) | 402944(95.01) | 6951(27.65) | 18185(72.34) |
| Age (years) mean (SD) | 60.22(6.87) | 56.36(8.11) | 67.79(13.28) | 59.93(16.73) |
| Female samples (%) | 8107(38.28) | 220047(54.60) | 3283(47.23) | 10274(56.40) |
| European samples (%) | 16636(78.56) | 339387(84.23) | 5457(78.51) | 15715(86.42) |
| Lipid lowering medication prescription (%) | 13433(63.44) | 56045(13.91) | 2197(31.61) | 2625(14.43) |
| TG concentration (mg/dL) median [IQR] | 171.83[125.09] | 129.58[95.66] | 125.00[93.00] | 97.00[70.00] |
| HDL-C concentration (mg/dL) mean (SD) | 46.04(12.44) | 56.51(14.71) | 49.78(17.82) | 58.88(19.80) |
| LDL-C concentration (mg/dL) mean (SD) | 111.59(34.17) | 138.98(33.04) | 90.74(35.88) | 104.08(35.89) |
| TC concentration (mg/dL) mean (SD) | 183.09(45.98) | 222.06(43.27) | 168.91(44.12) | 185.39(42.83) |

HLD-C–High-Density Lipoprotein Cholesterol; IQR–Inter quartile range; LDL-C–Low-Density Lipoprotein Cholesterol; MGBB–Mass General Brigham Biobank; SD–Standard Deviation; T2D –Type 2 Diabetes; TG–Triglycerides; TC–Total Cholesterol; UKB–UK Biobank.

(S3 Fig), where the minor allele (frequency 0.095) decreases natural log TG among those with T2D but yields no difference among those with non-T2D (T2D group: beta = -0.066, p-value = $3.9 \times 10^{-15}$; non-T2D group: beta = 0.002, p-value = 0.21; $p_{interaction}$ = $1.9 \times 10^{-11}$). We observed that the difference test ($Z_{Diff}$) identified this top significant lead variant (rs9274619: A; $Z_{diff}$ = -7.935) in the HLA locus as well. The mean raw TG measurements were significantly different between the reference and alternative genotypes in T2D samples (rs9274619(G/A or A/G): mean$_{diff}$ = 5.45 mg/dl; p-value = $1.26 \times 10^{-02}$; rs9274619(A/A): mean$_{diff}$ = 25.6 mg/dl; p-value = $3.78 \times 10^{-03}$). To evaluate the independence of the lead variant, we clumped the 478 genome-wide significant variants and performed conditional analyses. After clumping, we retained 8 individual signals, and rs3957148 was in strong LD with the lead variant. However, conditional analyses with all these eight clumped variants for rs9274619:A interaction with T2D status did not abrogate the lead variant's signal (S2 Table). Since BMI is associated with both TG and T2D, we used scaled BMI as an additional covariate in testing the association of

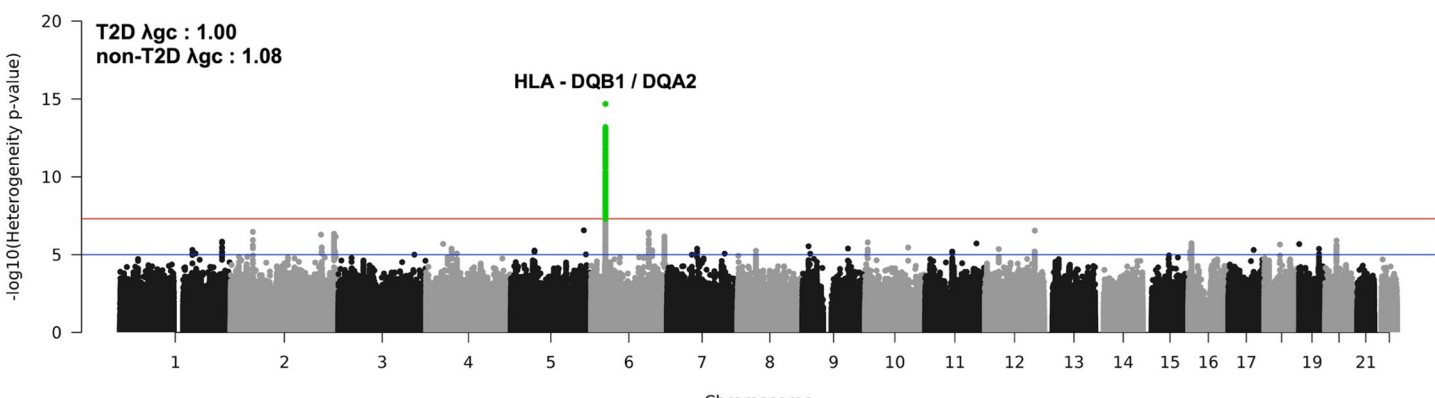

**Fig 1. Genome-wide heterogeneity between T2D strata.** Manhattan plot for heterogeneity p-values comparing T2D and non-T2D groups. Only one locus achieved genome-wide significance and the corresponding variants are colored in green. Lambda GC values from GWAS stratified by T2D status is shown in the figure. Red line: Genome-wide significance (p-value = $5 \times 10^{-8}$), Blue line: Suggestive significance (p-value = $1 \times 10^{-5}$). GWAS–Genome wide association studies; T2D –Type 2 Diabetes.

the identified lead variant with TG. We observed evidence for persistent albeit attenuated interaction between rs9274619:A and T2D after adjusting for BMI ($p_{interaction}$ = 2.9x10$^{-7}$). We observed a slight reduction in effects in the interaction model with T2D ($beta_{interaction}$ = -0.046), which shows that BMI has a confounding effect.

Individuals with T2D with lower TG concentrations were enriched for rs9274619:A (**Fig 2A**). Among individuals with normal TG (i.e., <150 mg/dL), rs9274619:A was associated with T2D by 1.23-fold (95% CI 1.15,1.29; p-value 2.8x10$^{-11}$). However, among individuals with TG > 450 mg/dl, rs9274619:A was not associated with T2D (OR 0.96, 95% CI 0.76–1.18; p-value 0.67). We replicated the findings in an independent cohort of 25,137 participants (6,951 T2D cases) of MGBB ($p_{heterogeneity}$ = 9.5x10$^{-3}$) (**Fig 2B**). Additionally, we evaluated the association of the lead variant interacting with T2D status with other lipids in discovery cohort (**S3 Table**). We observed a significant interaction between rs9274619:A and T2D on HDL-C ($p_{interaction}$ = 2.9x10$^{-8}$) with higher concentrations among those with T2D, and nominally greater reductions in LDL-C among those with T2D ($p_{interaction}$ = 6.0x10$^{-4}$).

We next bioinformatically prioritized the putative causal gene responsible for the T2D-dependent TG genetic association observed. Using T2D GWAS summary statistics PoPS prioritized the *HLA-DQB1* gene to be one of the top 20 genes along with other known TG genes such as *APOE*, *LPL* and *APOB*. However, *HLA-DQB1* was not prioritized in the non-T2D GWAS (**S4 Table**). Intersecting rs9274619:A with GTEx eQTL data for five different tissues (**Methods**) shows that the variant is an eQTL for multiple HLA genes including *HLA-DQB1* but more significantly for *HLA-DQA2* and *HLA-DRB6* (**S5 Table**). We curated all the eQTLs of the three HLA genes from GTEx database, T2D GWAS and correlated the Z-scores. eQTLs of all three HLA genes had a similar degree of correlations, but with opposite directions (**S4 Fig**). We further interrogated pQTL and mQTL data. rs9274619:A is a pQTL for *HLA-DQA2*

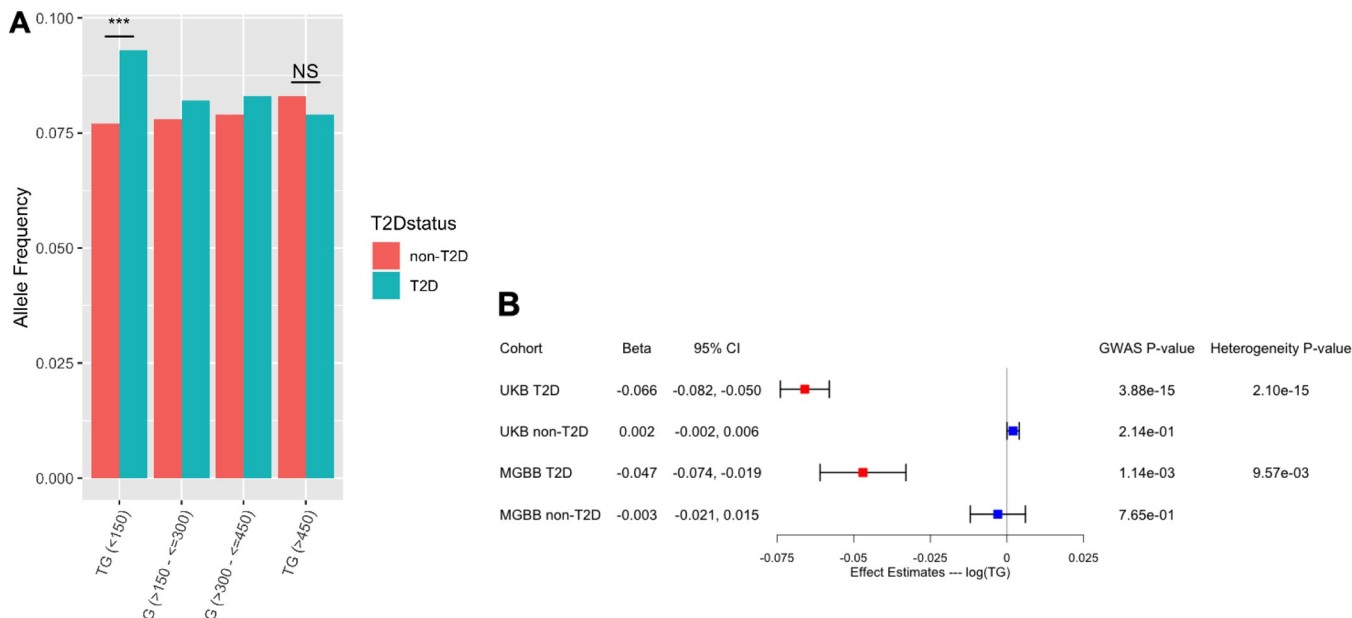

**Fig 2. *HLA-DQB1/DQA2* rs9274619:A significantly interacts with T2D on triglycerides.** A) Allele frequency of rs9274619:A in T2D and non-T2D samples grouped by raw TG values. Significance of samples proportions between the groups was assessed using Fisher's exact test for the lower and higher TG bins. B) GWAS and heterogeneity statistics of the lead variant rs9274619:A at the *HLA-DQB1/DQA2* locus from discovery (UKB) and replication (MGBB) cohorts based on T2D stratification. CI–Confidence Intervals; GWAS–Genome wide association studies; HLA–Human Leukocyte Antigens; MGBB–Mass General Brigham Biobank; NS–Non Significant; T2D –Type 2 Diabetes; TG–Triglycerides; UKB–UK Biobank.

(beta = 0.31; p-value = $6.2 \times 10^{-14}$) but it is an mQTL for multiple CpG regions at genome-wide significance. We identified 133 *cis*-associations and mapped the CpGs to Illumina Human-Methylation450 BeadChip (Illumina Inc., San Diego, USA) identification numbers (GEO data: GPL13534) to obtain the corresponding genes. Multiple HLA genes and other genes in chromosome 6 were identified (S6 Table). Gene prioritization using PoPS and QTL curation identified multiple HLA-genes (S5 Fig).

### rs9274619:A tags *HLA-DQB1*\*0302

Since the significant locus was at the HLA region, we correlated the rs9274619:A with 362 imputed HLA genotypes from 11 classes in the UKB. DQB1 and DQA1 were the most strongly correlated with rs9274619:A. Furthermore, DQB1_302 and DQA1_301 were most strongly correlated with rs9274619:A (DQB1_302: r = 0.95, $p_{correlation} < 3.83 \times 10^{-313}$; DQA1_301: r = 0.62, $p_{correlation} < 3.83 \times 10^{-313}$) (S6A Fig). We subsequently tested the interaction of all 362 HLA genotypes with T2D status on log(TG) as outcome. From this focused assessment of 362 HLA genotypes (S7 Table), 7 passed Bonferroni corrected significance ($0.05/362 = 1 \times 10^{-4}$) (S6B Fig). Consistent with our discovery and correlation analyses, only DQB1_302 had a genome-wide significant interaction ($p_{interaction} = 1.05 \times 10^{-9}$). Although rs9274619:A is associated with increased expression of *HLA-DQA2* gene in eQTL and pQTL analysis, alleles from these HLA types were not previously imputed in UKB. Since rs9274619:A located between both *HLA-DQB1* and *HLA-DQA2*, the variant could be a potential quantitative trait loci to both the genes.

Since the HLA locus is a strong predictor of type 1 diabetes (T1D), we used polygenic risk scores (PRS) from 67 variants previously associated with T1D as reported by Oram *et al.* [10], to exclude potential T1D cases in sensitivity analysis. First, we used the 67 SNPs from both HLA and non-HLA loci to create a genome-wide PRS for all the UKB samples. The top one percentile of the samples based on PRS were classified as T1D (N = 4242). Next, from the whole UKB samples, we removed any sample which identified as T1D by either ICD codes or PRS scores. With the remaining 18460 T2D cases, we observe a nominal interaction of the lead SNP with T2D ($beta_{interaction} = -0.026$; $p_{interaction} = 1.16 \times 10^{-02}$).

### Phenome-wide interaction analyses implicates multiple diabetes-related complications

We assessed the interactions between the rs9274619:A and T2D with 1567 disease conditions as outcomes (combination of incidence and prevalence) adjusted for all the covariates (age, $age^2$, sex, race, PC1-10). Using a Bonferroni correction ($0.05/1567 = 3.19 \times 10^{-5}$), 45 disease phenotypes exhibited significant interactions. The strongest interaction was for the concomitant diagnosis of type 1 diabetes (T1D) among those with T2D ($p_{interaction} < 1.72 \times 10^{-274}$) (S8 Table). We applied logistic regression models stratified by T2D status on the 45 significant phenotypes, while adjusting for all covariates as mentioned above (S9 Table). Multiple diabetes-related microvascular and macrovascular complications including hypoglycemia, retinopathy, polyneuropathy, angiopathy, atherosclerosis and osteomyelitis were significantly associated, with the T2D-specific TG-lowering rs9274619:A allele leading to increased risks (Fig 3). Interestingly, a recent investigation on Anti-neutrophil cytoplasmic antibody (ANCA)-associated vasculitides (AAV) by Dahlqvist *et al.* have identified rs9274619 as a lead variant for myeloperoxidase (MPO)-ANCA association [11], showing its importance in immune related vascular diseases. However, this allele was associated with reduced odds for obesity and related phenotypes.

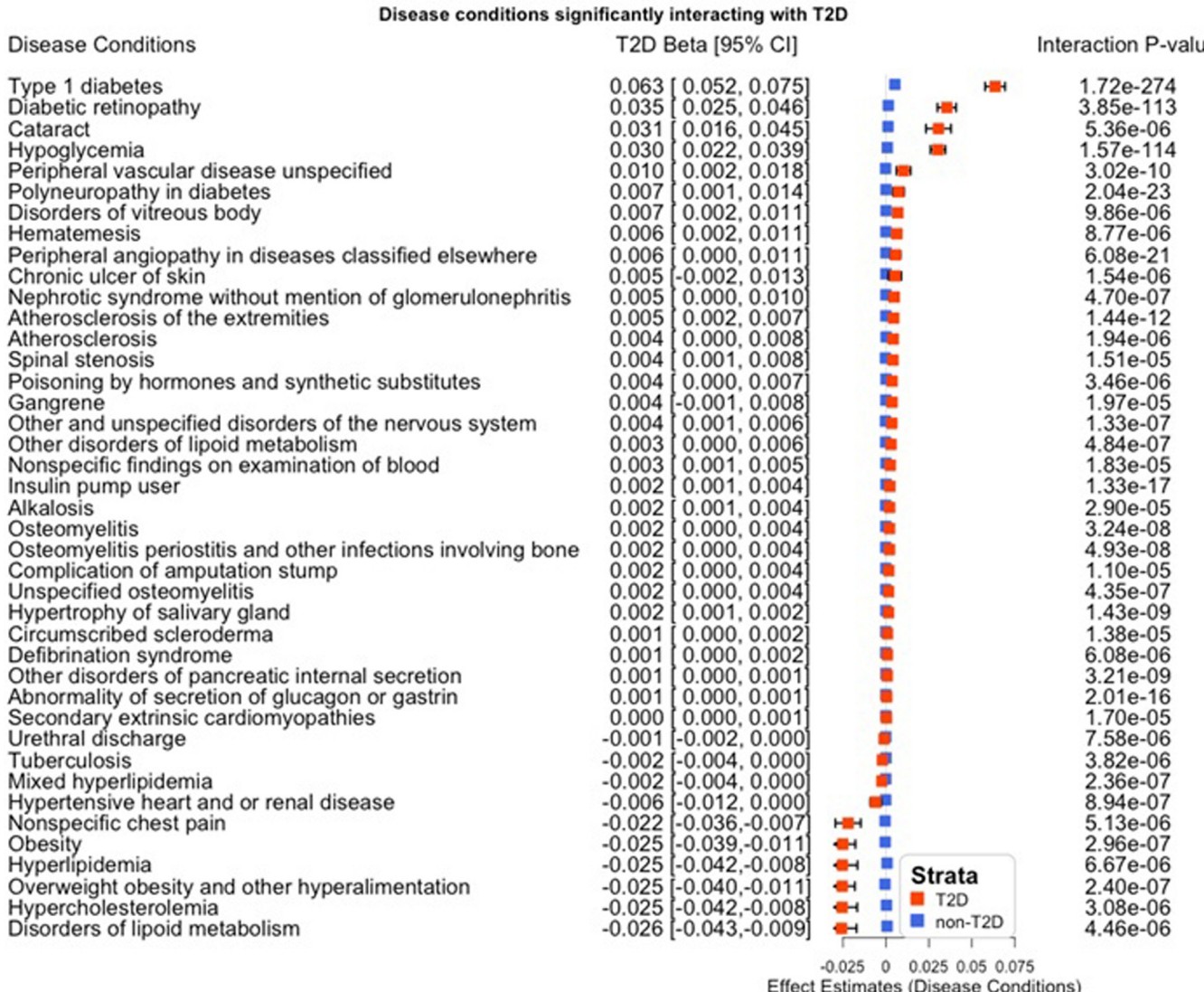

**Fig 3. Phenome wide association (PheWAS) of disease conditions stratified by T2D status.** Interaction between the rs9274619:A and T2D with multiple disease conditions as outcomes (combination of incidence and prevalence) was modeled while adjusting for all covariates (sex, age, age$^2$, race, PC1-10). Bonferroni corrected 45 significant phenotypes were stratified by T2D status and analyzed using logistic regression. The T2D effect estimates for rs9274619:A and the interaction p-value are documented, the disease conditions are ordered based on T2D beta. From the 45 disease conditions tested, highly correlated Type1 diabetic conditions were removed while plotting the figure. PC–Principal Components; PheWAS–Phenome Wide Association Studies; T2D –Type 2 Diabetes.

Since HLA-DQB1 is a GWAS locus for an overlap between T1D and T2D previously referred to as latent autoimmune diabetes in adults (LADA) [12], as well as T1D itself [13], we further explored the relationships between the rs9274619:A, T1D, T2D, and their respective interactions on TGs (**S10 Table**). The TG-lowering rs9274619:A was strongly associated with T1D after adjusting for T2D (p-value = 8.6x10$^{-113}$), but not significantly associated with T2D status after adjusting for T1D (p-value = 0.29). However, when assessing for interactions on TGs, there was still a significant interaction with T2D independent of T1D (beta$_{interaction}$ = -0.055; p$_{interaction}$ = 5.28x10$^{-9}$) and more strongly with T1D independently of T2D (beta$_{interaction}$ = -0.252; p$_{interaction}$ = 5.53x10$^{-49}$). Furthermore, we removed T1D samples and the interaction of

rs9274619:A with T2D on TG was nominally significant (beta$_{interaction}$ = -0.022; p$_{interaction}$ = 2.9x10$^{-2}$). We additionally checked the main effects of rs9274619:A for presumed LADA samples (N = 2580) in our dataset and observed a significant association (p-value = 8.37x10-41), showing that overlapping T1D and T2D samples could potentially contribute to the observed association of this locus.

Cousminer *et al.* reported four loci to be significantly associated with LADA [14], therefore we assessed T2D/TG interactions for these lead variants (S11 Table). None of the variants tested had a genome-wide significant interaction. The variant rs9273368 from *HLA-DQB1* (p$_{interaction}$ = 2.58x10$^{-5}$) was genome-wide significant in our heterogeneity analysis (p$_{heterogeneity}$ = 1.8x10$^{-8}$) and was in moderate LD with rs9274619:A (R$^2$ = 0.3). The four LADA loci were examined for interactions for additional diabetes-related phenotypes as noted in S12 Table.

## Metabolic characterizations of interactions with T2D

We secondarily explored the relationship between the rs9274619:A, interacting with T2D, and relationships with other metabolic features in UKB. Both the main effects and interaction models were adjusted for all the covariates (age, age$^2$, sex, race, PC1-10). Outcomes assessed included waist/hip ratio (WHR), body mass index (BMI), macronutrients from 24-hour dietary recall surveys, and 60 plasma biomarkers (S13 and S14 Tables). Several features were associated with the rs9274619:A itself, including increased eosinophil and neutrophil counts as well as hemoglobin A1c and C-reactive protein concentrations. We identified several outcomes that demonstrated differential association by T2D status in interaction testing (alpha 0.05/72 = 6.94x10$^{-4}$). The TG-lowering rs9274619:A allele was associated with greater concentrations for T2D vs non-T2D for hemoglobin A1c, sex hormone binding globulin, HDL-C, glucose, and apolipoprotein-A1 concentrations. However, the TG-lowering rs9274619:A allele was associated with reduced concentrations for T2D vs non-T2D for urate, BMI, WHR, and reticulocyte count (S13 and S14 Tables). Since HbA1c is a strong predictor of T2D, HbA1c is the most significant biomarker interacting with the lead SNP (S14 Table). When also using HbA1c > = 48 mmol/mol to define T2D samples as described by Young *et al.* [15], 15,180 samples were identified, with an additional 5212 samples to the previous T2D group. The interaction model of HbA1c-based only T2D with the lead SNP showed higher significance with TG (beta$_{interaction}$ = -0.159; p$_{interaction}$ = 1.26x10$^{-54}$). Next, we added the 5212 samples to our existing ICD10-based T2D groups and conducted a sensitivity analysis for the interaction. The addition of samples showed consistent significant interaction (beta$_{interaction}$ = -0.102; p$_{interaction}$ = 1.03x10$^{-35}$).

We further assessed lipid metabolomic data comprising 102,575 UKB samples (5200 T2D cases) and 249 metabolites for interactions. For each normalized metabolomic phenotype we analyzed the main effect of rs9274619:A and the interaction of rs9274619:A with T2D, with both models adjusted for all aforementioned covariates. Using a Bonferroni correction (0.05/249 = 2.01x10$^{-4}$), we identified 6 metabolomic features associated with the rs9274619:A and only 1 interaction with T2D (S15 and S16 Tables). With respect to the interaction detected, we observed that the average diameter for LDL particles among rs9274619:A carriers was greater for T2D versus non-T2D.

## Discussion

Independent GWAS studies have identified multiple loci strongly associated with TG and T2D separately [16, 17], and we now observe a variant tagging *HLA-DQB1/DQA2* associated with TGs only among those with T2D and not among those without T2D. Despite being associated with reduced TGs specifically in T2D, the lead variant is associated with greater diabetes-

related complications and an overlap with T1D. Previous studies have shown TG as an independent risk factor for cardiovascular diseases in T2D patients, and a predictor of T2D itself. Through a genome-wide stratified analysis, we now show that the HLA locus may triangulate some of these relationships by specifically influencing T2D-associated triglycerides.

Our study permits several conclusions regarding T2D pathogenesis. First, our study highlights heterogeneous metabolism among individuals with adult-onset diabetes. Hyperinsulinemia contributes to reduced hydrolysis and clearance of TG rich lipoproteins and thus persistence of these atherogenic lipoproteins toward heightened macrovascular risk [18]. TGs have more moderate associations with microvascular complications among diabetics [19]. Indeed, glycemic control is a more potent risk factor for microvascular complications. Here, we find that an immune-related locus (i.e., *HLA-DQB1/DQA2* genotype of major histocompatibility class II) linked to reduced TGs interestingly associates with greater microvascular versus macrovascular risk among individuals with T2D. Lipid homeostasis plays a key role in immune cells, where lipids are key constituents of major histocompatibility complex molecules and other cell membrane microdomains [20]. These observations highlight complementary roles of immune dysfunction and hyperinsulinemia in adult-onset diabetes pathogenesis.

Second, the distinct lipid pattern observed by HLA-DQB1*0302 genotype may reflect etiologically distinct subgroups of adult-onset diabetes. The HLA-DQB1*0302 has long been recognized as a very strong risk factor for T1D and tags the potent T1D DR4 risk haplotype [21–24]. Approximately 2–12% of adults diagnosed with type 2 diabetes have glutamic acid decarboxylase autoantibodies (GADA), thereby leading to the proposed term of latent autoimmune diabetes in adults (LADA) [25–27]. Such individuals are often classified as T2D because they typically do not initially require insulin. Indeed, a recent GWAS of LADA showed that *HLA-DQB1* was the most significant locus [14]. Recent data-driven approaches to cluster diabetes have grouped diabetes with GADA, traditionally classified as T1D or LADA, as severe autoimmune diabetes (SAID) [28]. Consistent with separate T1D and LADA analyses, the *HLA-DQB1* locus is significantly associated with SAID unlike with other diabetes subgroups [29]. Thus, the relatively reduced TG concentrations among adults classified as having T2D and the *HLA-DQB1* risk allele may reflect the lack of hypertriglyceridemia typically observed with more typical hyperinsulinemic T2D.

Third, TG concentrations among individuals diagnosed with T2D may help identify individuals with features more consistent with T1D. GADA testing only in adult-onset diabetics with normal TGs would optimize diagnostic yield. Furthermore, with increasingly available genotyping through expanding research testing and widely used direct-to-consumer approaches, HLA genotypes may further improve efficiency of testing. While large-scale randomized controlled trials for LADA are lacking, expert consensus recommend personalized management approaches deviating from conventional T2D and surveillance management [30]. Such approaches include biomarker-based surveillance of residual beta cell function and to determine insulin initiation.

A few limitations of our study deserve mention. First, causal gene prioritization through multiple methods did not converge on a single gene. Whether our observations reflect coordinated regulation merits further study. Based on the current results we were not able to elucidate the exact mechanism of TG lowering by HLA rs9274619:A in the context of T2D. Second, GADA or other islet autoantibodies and C-peptide levels are not present, despite using ICD codes and T1D polygenic risk score in sensitivity analyses, we cannot rule out the possibility of T1D-like features in some of the classified T2D cases. Third, UKB and MGBB are predominantly White and this may limit generalizability to other genetic backgrounds. Finally, Cochran's Q test is under powered to detect differences in heterogeneity of effect sizes and we

may have missed other loci with differences in effects. However, we also performed a difference test and found similar results to using the Cochran's Q test.

In conclusion, we observed that rs9274619:A linked to *HLA-DQB1/DQA2* is associated with reduced TGs only among adults with diabetes. Presence of this allele reflects an autoinflammatory subgroup of adult-onset diabetes most consistent with T1D, without characteristic hyperinsulinemia and thus relatively reduced TG concentrations. Among individuals classified as T2D, these individuals have greater risks for diabetes-associated complications.

## Materials and methods

### Study participants

We used the UK Biobank (UKB), which is a prospective population-based cohort composed of approximately 500,000 samples with rich phenotypic and genotypic information, as the discovery cohort [31]. UKB includes volunteer residents of the UK aged 40 to 69 years recruited during 2006–2010. The phenotypic information includes details on lifestyle, medical history, food habits, weight, height, body measurements, scans, blood routines, and electronic medical record (EMR) coded data. Out of 488,377 total individuals, we removed unconsented individuals and samples with >10% genotypic missingness thereby retaining 424,120 individuals for whom the TGs were also available.

We used data from the Mass General Brigham Biobank (MGBB), which comprises volunteer patients of the large Mass General Brigham Healthcare system in Massachusetts with greater than 105,000 participants, as replication [32]. In total 36,424 randomly selected individuals were genotyped using three versions of the Multi-Ethnic Genotyping Array (MEGA) Single-Nucleotide Polymorphism (SNP) array (Multiethnic Exome Global (Meg), Human multi-ethnic array (Mega), Expanded multi-ethnic genotyping array (Megex)). Out of 36,424 individuals, we retained 25,137 samples for whom T2D status and TG measurements were available for the current study.

**Ethics.** All UK Biobank participants gave written, informed consent per the UKB primary protocol. Secondary use of these data was approved by the Massachusetts General Hospital Institutional Review Board (protocol 2021P002228) and was facilitated through UKB application 7089. All MGB Biobank participants provided written, informed consent per the MGBB primary protocol. Secondary use of these data was approved by the Massachusetts General Hospital Institutional Review Board (protocol 2020P000904). All participants consent to broad use of their samples and data for research, no minor participants were included in this study.

### Phenotypes

In the UKB, we defined T2D based on self-reported status (data field 20002) and ICD10 codes E11:0–9 (data fields 41202, 41204, 40001, and 40002). The first instance of TG measurement (data field 30870) was defined as the primary lipid phenotype of interests. We also included other lipid levels as secondary outcomes: total cholesterol (TC) (data field 30690), low-density lipoprotein cholesterol (LDL-C) (data field 30780), and high-density lipoprotein cholesterol (HDL-C) (data field 30760). TG measurements were converted to mg/dL by multiplying mmol/L values by 88.57 and natural log transformed. TC, LDL-C and HDL-C values in mmol/L were converted to mg/dL by multiplying 38.67. When lipid-lowering medications were prescribed, TC measurements were divided by 0.8 and LDL-C by 0.7, as previously done [17]. All four lipid measurements were further inverse rank normalized to the residuals scaled by the standard deviation, where the model was adjusted for covariates (sex, age, age$^2$, PC1-10).

We curated multiple diseases for UKB samples into phecodes for PheWAS analysis. The PheWAS R package (version PheWAS_0.99.5–4) was used to map ICD codes to phecodes

based on the phecode map 1.2 and 1.2b1 from https://phewascatalog.org/ [33]. Codes that failed to map were excluded, which were relatively few and often procedural. Mapped codes were defined as multiple disease conditions and specified as incident or prevalent based on the time of sample collection. Next, we obtained the secondary phenotypes, which included waist circumference (data field 48), hip circumference (data field 49), body mass index (BMI) (data field 23104) and 24-hour diet recall (data field 110001) for downstream analysis. Additionally, we included blood biochemistry (category id 18518) and count (category id 9081) measures. These phenotypes were normalized to a mean 0 and standard deviation 1 for analysis. We obtained the NMR metabolic biomarkers generated by Nightingale Health (Helsinki, Finland) from the first tranche of 249 metabolic biomarkers in 118,032 UKB participants [34]. We included 102,528 samples that intersected with the discovery cohort and each of the metabolites were inverse rank normalized and regressed against the covariates (age, age$^2$, sex, race, PC1-10). The residuals were used as our phenotypes in analyses.

In MGBB, electronic health record (EHR) data were used to define incident and prevalent cases based on enrollment date and ICD-9/ICD-10 codes on clinical phenotype definitions from phecode groups [35, 36], where samples with phecode 250.2X were defined as T2D in our study. Similarly, lipid test results, medication information, demographic status of genotyped samples were curated from EHR records. LDL-C was measured directly or calculated using Friedewald equation when TG were <400 mg/dL, all lipid measurements were in mg/dL units. The lipid measurements closest to sequencing date was curated. A sample was defined as on statin medication, if statin treatment was prescribed within the last one year of the sequencing date. We performed phenotype harmonization and normalization for the validation data as described above.

## Genotypes

Genetic data from 488,377 UKB samples were assayed using two similar genotyping arrays from Affymetrix (Santa Clara, CA): i) Applied Biosystems UK BiLEVE Axiom Array ii) Applied Biosystems UK Biobank Axiom Array. 49,950 participants with 807,411 markers were genotyped at using the Applied Biosystems UK BiLEVE Axiom Array and 438,427 participants with 825,927 markers were genotyped using the closely related Applied Biosystems UK Biobank Axiom Array. Both arrays shared 95% of marker content and the UK Biobank Axiom array was chosen to capture genome-wide genetic variation (single nucleotide polymorphism (SNPs) and short insertions and deletions (indels)) [31]. The imputation from the UKB array-derived genotypes was performed using merged UK10K and 1000 Genomes phase 3 reference panels [37] and was combined to the Haplotype Reference Consortium (HRC) [38] panel using IMPUTE4 program (https://jmarchini.org/software/) as implemented in IMPUTE2 [39]. We obtained the UKB imputed human leukocyte antigen (HLA) genotypes (data field 22182) composed of classical allelic variation of 11 HLA types (A, B, C, DRB5, DRB4, DRB3, DRB1, DQB1, DQA1, DPB1, DPA1). HLA imputation from allele pairs was performed using HLA*IMP:02 in the UKB, as previously described [31]. Genotypic data in MGBB cohort was generated using three different arrays (Multiethnic Exome Global [MEG], Human multi-ethnic array [MEGA], Expanded multi-ethnic genotyping array [MEGEX]) from Illumina (San Diego, CA).

## Statistical analysis

We performed genome wide association analysis (GWAS) stratified based on T2D status. We first performed quality control (QC) of the full UKB dataset regardless of T2D status by applying additional filters, including minor allele frequency (MAF) < 1%, Hardy-Weinberg equilibrium p-value not exceeding 1x10$^{-15}$ and genotype missingness > 10% to filter variants, and

sample-level genotype missingness > 10%. The QC-passed dataset was used to create NULL model with sex, age, age$^2$, genotype array, race and PC1-10 as covariates. We employed REGENIE with leave-one-out-cross-validation (LOOCV) [40] approach adjusted for covariates stated above to perform GWAS on UKB imputed data with minor allele count (MAC) 20 in both T2D and non-T2D samples, independently. We annotated the genomic risk-loci from GWAS summary statistics using FUMA [41].

We tested for differences in effect estimates per genotype by T2D status using Cochran's Q-test for heterogeneity in the METAL package [42]. For the genome-wide heterogeneity assessment, we used the conventional alpha threshold of $5\times10^{-8}$ to assign statistical significance accounting for multiple-hypothesis testing. We validated the outcomes from the Cochran's Q-test using the Difference Test [43]. We clumped the genome significant variants using plink [44] (p1: 5e-08; r2:0.5; kb:250). Significant lead variant in the discovery dataset were replicated at an alpha threshold of 0.05 accounting for the single SNP assessed.

The significant and replicated variant (lead variant) was pursued for further downstream analysis. Given its genomic location, we correlated the UKB imputed classical HLA genotypes with the significant lead variant using corplot R-package (method-pearson; version-0.90). We applied sample and variant quality measures to the initial data (—geno 0.05;—mind 0.05;—hwe 1E-06) and calculated PRS using plink [44]. We performed regression-based interaction analyses using the model where adiposity-related, diet-related and other blood biomarker phenotypes were separately analyzed with the lead variant along with T2D status. The regression analysis (main and interaction model) was carried out in R, adjusting for sex, age, age$^2$, genotype array, race, and PC1-10 as covariates. Bonferroni corrected alpha threshold of 0.05/number of tests was considered statistically significant for these analyses. Fisher's exact test was performed to test the significance of sample proportions among group of samples.

We implemented the Polygenic Priority Score (PoPS) enrichment method [45] for gene prioritization with the GWAS summary statistics. PoPS integrates multiple public bulk and single-cell expression datasets, protein-protein interaction and pathway databases to implement enrichment analysis using MAGMA [46] based gene association scores to identify top list of genes functionally linked to the phenotype of interest. We complementarily performed quantitative trait locus (QTLs) interrogations using multiple publicly available datasets. We downloaded expression quantitative trait locus (eQTLs) data from GTEx (v8_eQTL_all_associations) database (https://gtexportal.org/home/datasets) and curated significant hits (p-value < $5\times10^{-8}$) for the lead SNP from 5 different tissues relevant to diabetes, lipids, and inflammation (i.e., Liver, Adipose Subcutaneous, Adipose Visceral Omentum, Whole Blood, and Pancreas). We utilized protein quantitative trait loci (pQTL) data in blood from the INTERVAL study [47] and GoDMC database [48] for methylation quantitative trait loci (mQTL) in blood to curate pQTLs and mQTLs related to the lead variant (p-value < $5\times10^{-8}$).

## Supporting information

**S1 Fig. Overall study schematic.** We carried out stratified GWAS on UKB discovery cohort based on T2 status, using Regenie LOOCV models adjusting for age, age$^2$, sex, race, and PC1-10. We implemented heterogeneity analysis to identify loci that was differentially associated between the two strata. Out of the 67M variants analyzed, only one locus achieved genome-wide significance. We replicated the significant locus using MGBB, an independent cohort, and further analyzed the lead variant using various secondary analysis in the discovery cohort. GWAS–Genome wide association; LOOCV—leave-one-out-cross-validation; MGBB–Mass General Brigham Biobank; T2D –Type 2 Diabetes; UKB–UK Biobank.
(TIF)

**S2 Fig. Manhattan (MH) plots for T2D stratified GWAS.** A) MH plots for T2D GWAS B) MH plots for non-T2D GWAS. Genes near to the most significant lead variant in each loci are documented, full list of lead SNPs are tabulated in S1 Table. Red line: Genome significance (p-value = 5x10$^{-8}$), Blue line: Suggestive significance (p-value = 1x10$^{-5}$). GWAS–Genome wide association studies; MH–Manhattan; T2D –Type 2 Diabetes.
(ZIP)

**S3 Fig. *HLA-DQB1/DQA2* lead variant locus in T2D and non-T2D strata.** A) Locus zoom plot for *HLA-DQB1/DQA2* loci in T2D strata. B) Locus zoom plot for *HLA-DQB1/DQA2* loci in non-T2D strata. X-axis defines the genomic position where variants +/-500 kb on either side of rs9274619—chr6:32635954:G:A (grc37) is mapped on the genome. The variants are colored based on the r$^2$ with the lead variant and the genes are mapped based on their genomic position. Y-axis is the -log10(p-values) from the respective strata and the scale of y-axis is different between the two plots. HLA–Human Leukocyte Antigen; T2D –Type 2 Diabetes.
(ZIP)

**S4 Fig. Correlation between GTEx and GWAS Z-scores of eQTLs from *HLA-DQB1, HLA-DQA2* and *HLA-DRB6*.** Z-scores were calculated from T2D GWAS and GTEx (version 8) summary statistics for all the eQTLs for the three genes. Pearson correlation coefficient was calculated, and scatter plots were generated for eQTL data from five different tissues. Most of the TG lowering variants increases the expression of *HLA-DQA2/HLA-DRB6*, whereas decreases the expression of *HLA-DQB1*.
(ZIP)

**S5 Fig. Gene prioritization using PoPS and QTL mining.** PoPS method was used to prioritize genes using GWAS summary statistics from both T2D and non-T2D stratum. Multiple HLA genes were prioritized, where *HLA-DQB1* topped the list. eQTL, pQTL and mQTL curation of rs9274619:A from various public repositories mapped the lead variant to multiple HLA-genes, where *HLA-DQA2* was identified by all three QTL searches.
(TIF)

**S6 Fig. Imputed HLA alleles and its correlation with variant of interest—rs9274619:A.** A) Correlation between rs9274619:A and HLA alleles in DQB1 and DQA1 class. DQB1-302 is the most strongly correlated allele. B) Forest plot showing the different alleles that passed the Bonferroni correction on interacting with T2D with log(TG) as outcome, the model was adjusted age, age$^2$, sex, race, PC1-10 and rs9274619:A. DQB1-302 allele is the only allele with significant interaction with T2D and highly correlated to rs9274619:A (mapped in red). HLA–Human Leukocyte Antigen; T2D –Type 2 Diabetes; TG–Triglycerides; VOI–Variant of interest.
(ZIP)

**S1 Table. Genomic risk loci identified from GWAS stratified by T2D status in UKB.** 19 and 315 risk loci were identified by FUMA on T2D and non-T2D GWAS respectively. Significant loci and lead variants identified from FUMA analysis for each risk loci are documented. GWAS–Genome wide association studies; UKB–UK Biobank; T2D –Type 2 Diabetes.
(XLSX)

**S2 Table. List of variants identified after clumping the 478 SNPs in the chromosome 6 loci.** The r2 of the clumped variant with the lead SNP and their respective summary data are documented. Each clumped SNP was additionally used to carry out conational analysis with the lead SNP interaction model and the summary statistics are documented.
(XLSX)

**S3 Table. Summary statistics of lead SNP with lipids.** Summary statistics from linear regression model where rs9274619:A interacting with T2D is documented for log(TG) as outcome for the 3 lipids including LDL-C, HDL-C and TC. *HLA-DQB1/DQA2* is enriched only in T2D and not in non-T2D. HLD-C–High-Density Lipoprotein Cholesterol; LDL-C–Low-Density Lipoprotein Cholesterol; T2D –Type 2 Diabetes; TG–Triglycerides; TC–Total Cholesterol. (XLSX)

**S4 Table. Gene prioritization scores.** PoPS enrichment scores for the top100 genes from T2D and nonT2D GWAS summary statistics. The Genes are ordered based on PoPS scores within each strata. T2D –Type 2 Diabetes. (XLSX)

**S5 Table. Summary statistics for curated eQTLs.** eQTL–gene pairs for rs9274619:A was curated from GTEx data and genome significant hits were obtained (p-value = $5 \times 10^{-8}$) from 5 different tissues. Each associated gene is tabulated and ordered based on GTEx p-value. eQTL–Expression Quantitative Trait Locus. (XLSX)

**S6 Table. Summary statistics for curated mQTLs.** mQTL–CpG pairs for rs9274619:A and the corresponding genes that regulate the CpG regions was curated from GoDMC database where the genome significant hits were obtained (p-value = $5 \times 10^{-8}$) from the cis-associations. The data is ordered based on p-value and the corresponding CpG island regions were curated from Illumina HumanMethylation450 BeadChip resource. mQTL–Methylation Quantitative Trait Locus. (XLSX)

**S7 Table. Summary statistics for curated HLA-alleles.** Summary statistics for linear regression HLA interaction model with T2D status adjusted for age, age^2, sex, race, PC1-10 and rs9274619:A, where the outcome was normalized log(TG). The correlation coefficient of each HLA allele to rs9274619:A is documented. The HLA alleles are ordered based on their significance, out of the total 362 HLA alleles 331 alleles had interaction summary statistics. PC–Principal Components; T2D –Type 2 Diabetes; TG–Triglycerides. (XLSX)

**S8 Table. Summary statistics from PheWAS.** Summary statistics for logistic regression model for PheWAS where disease conditions used as outcomes, adjusted for age, age^2, sex, race and PC1-10. The effects from the rs9274619:A interacting with T2D status is documented and the disease conditions are ordered based on significance. PC–Principal Components; PheWAS–Phenome Wide Association Studies; T2D –Type 2 Diabetes. (XLSX)

**S9 Table. Summary statistics from significant disease associations.** The 45 significant disease conditions identified using interaction model were analyzed using logistic regression, adjusted for age, age^2, sex, race and PC1-10 and stratified by T2D status. The summary statistics from T2D and non-T2D models are tabulated, where the phenotypes are ordered based on T2D beta. PC–Principal Components; T2D –Type 2 Diabetes. (XLSX)

**S10 Table. Summary statistics from T1D and LADA linear models.** Multiple linear main and interaction models used to validate the influence of T1D and LADA samples. LADA–Latent Autoimmune Diabetes in Adults; T1D –Type 1 Diabetes. (XLSX)

**S11 Table. Summary statistics from LADA interaction models.** Interaction model between the four LADA loci and T2D with TG as outcome. LADA–Latent Autoimmune Diabetes in Adults; T2D –Type 2 Diabetes; TG–Triglycerides.
(XLSX)

**S12 Table. Significant disease phenotypes with LADA loci.** Interaction p-values between the four LADA loci and T2D with T1D related and hyperlipidemia related phenotypes as outcomes. The top 5 and bottom 5 phenotypes significantly associated with rs9274619:A were selected (Fig 3). All four loci have significant interaction with diabetes related diseases, but not with obesity related phenotypes. LADA–Latent Autoimmune Diabetes in Adults; T1D –Type 1 Diabetes; T2D –Type 2 Diabetes.
(XLSX)

**S13 Table. Summary statistics from linear models with biomarkers.** Summary statistics for linear regression main model with fat, diet and biomarkers phenotypes as outcomes, adjusted for age, age^2, sex, race and PC1-10. The phenotypes are orders based on p-value. PC–Principal Components.
(XLSX)

**S14 Table. Summary statistics from interaction models with biomarkers.** Summary statistics for linear interaction model with fat, diet and biomarkers phenotypes as outcomes, adjusted for age, age^2, sex, race and PC1-10. The phenotypes are orders based on p-value. PC–Principal Components.
(XLSX)

**S15 Table. Summary statistics from linear models with metabolites.** Summary statistics for linear regression main model with metabolomic phenotype as outcomes, adjusted for age, age^2, sex, race and PC1-10. The effects from rs9274619:A are documented and the metabolomes are ordered based on significance. PC–Principal Components.
(XLSX)

**S16 Table. Summary statistics from interaction models with metabolites.** Summary statistics for linear regression interaction model with metabolomic phenotype as outcomes, adjusted for age, age^2, sex, race and PC1-10. The effects from rs9274619:A interacting with T2D status is documented and the metabolomes are ordered based on significance. PC–Principal Components; T2D –Type 2 Diabetes.
(XLSX)

## Acknowledgments

We thank all the participants from UKB and MGBB.

## Author Contributions

**Conceptualization:** Stephen S. Rich, Gina M. Peloso, Pradeep Natarajan.

**Data curation:** Margaret Sunitha Selvaraj, Kaavya Paruchuri, Sara Haidermota, Rachel Bernardo.

**Formal analysis:** Margaret Sunitha Selvaraj, Pradeep Natarajan.

**Funding acquisition:** Pradeep Natarajan.

**Investigation:** Margaret Sunitha Selvaraj.

**Methodology:** Margaret Sunitha Selvaraj.

**Project administration:** Margaret Sunitha Selvaraj, Gina M. Peloso.

**Software:** Margaret Sunitha Selvaraj.

**Supervision:** Stephen S. Rich, Gina M. Peloso, Pradeep Natarajan.

**Validation:** Margaret Sunitha Selvaraj.

**Visualization:** Margaret Sunitha Selvaraj.

**Writing – original draft:** Margaret Sunitha Selvaraj, Stephen S. Rich, Gina M. Peloso, Pradeep Natarajan.

**Writing – review & editing:** Margaret Sunitha Selvaraj, Stephen S. Rich, Gina M. Peloso, Pradeep Natarajan.

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
