## [Decision Letter · Decision Letter 0]

21 Jul 2022

PONE-D-22-11074Genome-wide Discovery for Diabetes-Dependent Triglycerides-Associated LociPLOS ONE

Dear Dr. Natarajan,

Thank you for submitting your manuscript to PLOS ONE. After careful consideration, we feel that it has merit but does not fully meet PLOS ONE’s publication criteria as it currently stands. Therefore, we invite you to submit a revised version of the manuscript that addresses the points raised during the review process.

The manuscript is based on very impressive datasets, and integrates both genetic associations and expression data. The reviewers were generally positive and notice the strengths of both dataset and analyses approach. The DQB1*03:02 SNP for T2D and triglycerides is particularly interesting.

Their main concerns were the following.

1.    Could the association be with DQB1*03:02 SNP be caused by type 1 diabetes patients being intermixed with the T2D patients? (rev 2 and 3). Can this be addressed by analysing certain subsets of the data?

2.    Elaborate on the relationship of SNP, TG(endophenotype) and T2D and T1D. Rev 2. “from previous studies, seems TG was an independent risk factor of cardiovascular diseases in T2D patients, and a predictor of T2D itself. The authors may discuss the difference of the association in T2D and non-T2D subjects.”

3.    Is the lead SNP independent or not? Rev 3. “The authors need to provide further analysis that the present high risk SNP is not in linkage disequilibrium with other genetic factors.” This can be done with conditional analyses. https://www.ncbi.nlm.nih.gov/pmc/articles/PMC4572002/

4.    It would be interesting, though not required by journal to address the question. Is there a “… relationship between the HLA-DQB1*03:02 SNP and TG levels in the type 1 diabetes subjects in the UKBB “?

5.    Explain in the manuscript how other researchers can gain access to the data, note the PLOS one data policy.

We look forward to receiving your revised manuscript.

Kind regards,

Arnar Palsson, Ph.D.

Academic Editor

PLOS ONE

Journal Requirements:

P.N. is supported by grants from the National Institutes of Health (R01HL142711, R01HL148050, R01HL151283, R01HL127564, R01HL148565, R01HL151152, R01DK125782), Fondation Leducq (TNE-18CVD04), and Massachusetts General Hospital (Fireman Chair). GMP is supported by NIH grants R01HL127564 and R01HL142711.

We thank all the participants from UKB and MGBB. P.N. is supported by grants from the National Institutes of Health (R01HL142711, R01HL148050, R01HL151283, R01HL127564, R01HL148565, R01HL151152, R01DK125782), Fondation Leducq (TNE-18CVD04), and Massachusetts General Hospital (Fireman Chair). GMP is supported by NIH grants R01HL127564 and R01HL142711.

However, funding information should not appear in the Acknowledgments section or other areas of your manuscript. We will only publish funding information present in the Funding Statement section of the online submission form. 

P.N. is supported by grants from the National Institutes of Health (R01HL142711, R01HL148050, R01HL151283, R01HL127564, R01HL148565, R01HL151152, R01DK125782), Fondation Leducq (TNE-18CVD04), and Massachusetts General Hospital (Fireman Chair). GMP is supported by NIH grants R01HL127564 and R01HL142711.

P.N. reports grants from Amgen, Apple, AstraZeneca, Boston Scientific, and Novartis, personal fees from Apple, AstraZeneca, Blackstone Life Sciences, Foresite Labs, Genentech / Roche, Novartis, and TenSixteen Bio, equity in geneXwell, and TenSixteen Bio, co-founder of TenSixteen Bio, and spousal employment at Vertex, all unrelated to the present work. 

We note that you received funding from a commercial source: Amgen, Apple, AstraZeneca, Boston Scientific, and Novartis, personal fees from Apple, AstraZeneca, Blackstone Life Sciences, Foresite Labs, Genentech / Roche, Novartis, and TenSixteen Bio, equity in geneXwell, and TenSixteen Bio, co-founder of TenSixteen Bio, and spousal employment at Vertex

Additional Editor Comments:

The manuscript is based on very impressive datasets, and integrates both genetic associations and expression data. The reviewers were generally positive and notice the strengths of both dataset and analyses approach. The DQB1*03:02 SNP for T2D and triglycerides is particularly interesting.

Their main concerns were the following.

1. Could the association be with DQB1*03:02 SNP be caused by type 1 diabetes patients being intermixed with the T2D patients? (rev 2 and 3). Can this be addressed by analysing certain subsets of the data?

2. Elaborate on the relationship of SNP, TG(endophenotype) and T2D and T1D. Rev 2. “from previous studies, seems TG was an independent risk factor of cardiovascular diseases in T2D patients, and a predictor of T2D itself. The authors may discuss the difference of the association in T2D and non-T2D subjects.”

3. Is the lead SNP independent or not? Rev 3. “The authors need to provide further analysis that the present high risk SNP is not in linkage disequilibrium with other genetic factors.” This can be done with conditional analyses. https://www.ncbi.nlm.nih.gov/pmc/articles/PMC4572002/

4. It would be interesting, though not required by journal to address the question. Is there a “… relationship between the HLA-DQB1*03:02 SNP and TG levels in the type 1 diabetes subjects in the UKBB “?

5. Explain in the manuscript how other researchers can gain access to the data, note the PLOS one data policy.

Reviewers' comments:

Reviewer's Responses to Questions

**Comments to the Author**

1. Is the manuscript technically sound, and do the data support the conclusions?

Reviewer #1: Yes

Reviewer #2: Yes

Reviewer #3: Yes

2. Has the statistical analysis been performed appropriately and rigorously? 

Reviewer #1: Yes

Reviewer #2: Yes

Reviewer #3: Yes

3. Have the authors made all data underlying the findings in their manuscript fully available?

Reviewer #1: Yes

Reviewer #2: Yes

Reviewer #3: No

4. Is the manuscript presented in an intelligible fashion and written in standard English?

Reviewer #1: Yes

Reviewer #2: Yes

Reviewer #3: Yes

5. Review Comments to the Author

Reviewer #1: Dear Author,

I strongly recommend this publication for the following perspectives. First, the large cohort sample sizes for both the cases and controls increases the statistical power for detection of variants with small effect sizes, a vital limitation in analysis of complex genetic traits, were the genetic effect is due to multiple variants with small effect sizes. Second, adjustment for BMI as a potential confounder is of high relevance giving the precision of the statistical model. Third, the implementation of Cochran statistical method in METAL, is highly recommended in conduction meta-analysis of GWAS.

Fourth, linking the GWAS results to the gene expression in different tissues associated with lipid metabolism strongly augment and support the finding. Fifth, even though identification of genetic association of T2D with the well-known T1D HLA locus adds to the complexity of the phenotype yet it could be of valuable information reflecting the protective role of immune system in this category of T2D patients. Overall, great research and work, well written, it adds value for understanding the heterogeneity of T2D supporting the fact that there are underlying subtypes and identifying the role of immune system in T2D not only T1D and diabetic complications.

Reviewer #2: Selvaraj et al. performed genome-wide association and interaction studies among UKB and MGBB subjects, they found that the HLA-DQB1 locus was associated with lower triglyceride levels in T2D patients, but not in non-T2D subjects.

Several issues need to be addressed before acceptance for publication: 1) The affection status of T2D in UKB subjects was based on self-reported data, the incidence of T2D (5.0%) in UKB participants seems lower than general population, given the mean age of 56.6 years. Should fasting glucose and HbA1c be considered for the diagnosis of T2D? 2) The "concomitant diagnosis of T1D among those with T2D" is confusing. It is highly unlikely that a subject had both T1D and T2D, however, it could be an issue since affection statuses were self-reported. Indeed, the HLA-DQB1 locus was only associated with T1D, not with T2D, in previous GWASs in large cohorts. Since no information of LADA diagnosis was available in UKB, testing LADA loci in UKB (the number of LADA patients was limited) provided little information. 3) It is interesting to know that the HLA-DQB1 polymorphisms were associated with higher HbA1c and glucose levels, higher risks of diabetic marco- and microvascular complications, but lower TG, urate, and body weight phenotypes in T2D subjects. From previous studies, seems TG was an independent risk factor of cardiovascular diseases in T2D patients, and a predictor of T2D itself. The authors may discuss the difference of the association in T2D and non-T2D subjects.

Reviewer #3: 1. The study suffers from the habit of classifying the bulk of diabetes patients above 20-25 years of age as type 2 diabetes. The diagnosis is diabetes but the classification is type 2 diabetes based on loose clinical criteria. The strong association between low levels of TG and the A variant in the HLA-DQ B1*03:02 is a distinct example that the authors should discuss.

2. The authors should also discuss the lack of information on GADA or other islet autoantibodies, c-peptide levels as well as HbA1c and the weakness that most of the diabetes classification is self-reported.

3. Genetic Risk Scores (GRS) for autoimmune type 1 diabetes has been developed using the UKBB (Oram and others) and these data should be run in parallel with the present analysis. It may resolve the many issues of re-classifying the patients with autoimmune type 1 diabetes rather than maintaining that they are type 2 diabetes patients.

4. The region around the reported DQB1*03:02 SNP should be further analysed to provide a heat-map of the region to indicate that the particular SNP reported has the highest risk, or p-value, compared to neighbouring SNPs. The authors need to provide further analysis that the present high risk SNP is not in linkage disequilibrium with other genetic factors.

5. There are type 1 diabetes patients (usually more clearly defined when self-reported) in the UKBB. What is the risk of the present DQB1*03:02 SNP for type 1 diabetes? Is there data to indicate that this particular SNP marked low level TG also in type 1 diabetes patients?

6. PLOS authors have the option to publish the peer review history of their article (what does this mean?). If published, this will include your full peer review and any attached files.

Reviewer #1: **Yes: **Dina Mansour Aly

Reviewer #2: No

Reviewer #3: **Yes: **Åke Lernmark

---

## [Author Response · Author response to Decision Letter 0]

2 Sep 2022

EDITORS COMMENT:

The manuscript is based on very impressive datasets, and integrates both genetic associations and expression data. The reviewers were generally positive and notice the strengths of both dataset and analyses approach. The DQB1*03:02 SNP for T2D and triglycerides is particularly interesting.

Their main concerns were the following.

1. Could the association be with DQB1*03:02 SNP be caused by type 1 diabetes patients being intermixed with the T2D patients? (rev 2 and 3). Can this be addressed by analysing certain subsets of the data?

Author Response:

Thank you for this comment. We observed that rs9274619:A was more significantly associated with T1D (p-value 1.29x10-124) compared to T2D (p-value 9.69x10-14) as previously shown in Supplementary Table 10. We agree that sample intermixing could be one of the possible reasons. The concomitant diagnosis of T2D along with T1D is an active area of research with overlap often termed as latent autoimmune diabetes in adults (LADA), which could also be a potential explanation for the association at this locus. We have discussed this in the manuscript from lines 178-196.

To add clarity of the LADA specific SNP interaction we have added a sentence discussing the overlapping samples from both T1D and T2D forming the LADA group in the results section.

Text in the manuscript:

We additionally checked the effect of rs9274619:A for LADA samples (N=2580) in our dataset and observed significant association (p-value=8.37x10-41), showing that overlapping T1D and T2D samples could potentially drive the association of this loci.

Additionally, we subsetted samples based on T1D polygenic risk score (PRS) from Oram et al as suggested by Reviewer 3. Briefly, we used 67 SNPs to calculate a T1D PRS in UKB. Based on 99th percentile of the PRS, we identified and removed potential T1D samples (N= 4242) in UKB. Next, we removed residual T1D samples identified based on ICD codes as well. Finally, we show that the interaction model of the lead SNP with T2D is still nominally significant for TG (beta= -0.0261; p-value= 0.01). Through this analysis we show that even after the removal of potential T1D samples based on validated 67 SNPs of T1D GRS (Oram et al), the signal from the lead SNP rs9274619:A is nominally significant. While we cannot rule out the possibility of residual T1D case mixing, these results remain consistent with an interaction for T2D. We have added this analysis to the results section.

Text in the manuscript:

Since the HLA locus is a strong predictor of type 1 diabetes (T1D), we used polygenic risk scores (PRS) from 67 variants previously associated with T1D as reported by Oram et al(10), to exclude potential T1D cases in sensitivity analysis. First, we used the 67 SNPs from both HLA and non-HLA loci to create a genome-wide PRS for all the UKB samples. The top one percentile of the samples based on PRS were classified as T1D (N=4242). Next, from the whole UKB samples, we removed any sample which identified as T1D by either ICD codes or PRS scores. With the remaining 18460 T2D cases, we observe a nominal interaction of the lead SNP with T2D (betainteraction=-0.026; pinteraction=1.16x10-02). 

2. Elaborate on the relationship of SNP, TG(endophenotype) and T2D and T1D. Rev 2. “from previous studies, seems TG was an independent risk factor of cardiovascular diseases in T2D patients, and a predictor of T2D itself. The authors may discuss the difference of the association in T2D and non-T2D subjects.”

Author Response:

Thank you for this comment. It is true that TG is an independent risk factor for cardiovascular diseases and a predictor of T2D. In this study, we performed a stratified analysis to capture the heterogeneity by genetic variants by T2D status. Though the independent GWAS of T2D and non-T2D groups identified significant loci, from our analysis we were able to show that genome-wide significant heterogeneity was present at the chromosome 6 locus. This may have bearing on pathways that may specifically influence T2D-associated cardiovascular disease with less relevance outside the context of T2D. The chromosome 6 HLA locus is a strong locus for T1D odds but given the heterogeneity in the associations between T2D and non-T2D, we show that this locus has the strongest signal to differentiate T2D to non-T2D. Our analysis could identify hidden signals between the stratified groups of samples and could pave way to identify new loci. We have added a sentence in the discussion section of the manuscript addressing the question.

Text in the manuscript:

Previous studies have shown TG as an independent risk factor for cardiovascular diseases in T2D patients, and a predictor of T2D itself. Through a genome-wide stratified analysis, we now show that the HLA locus may triangulate some of these relationships by specifically influencing T2D-associated triglycerides.

3. Is the lead SNP independent or not? Rev 3. “The authors need to provide further analysis that the present high risk SNP is not in linkage disequilibrium with other genetic factors.” This can be done with conditional analyses. https://www.ncbi.nlm.nih.gov/pmc/articles/PMC4572002/

Author Response:

Thank you for this comment and suggestion. We have investigated the locus in detail showing that the association from lead SNP (rs9274619:A) is independent of other variants in the region. To demonstrate this, we performed variant clumping with all 478 genome-wide significant SNPs in chr6p21.32 loci and identified 8 independent signals. We tabulated the R2 for these SNPs (using 1000G reference panel) with the lead SNP analyzed in this study and provided summary statistics for these clumped SNPs as Supplementary Table 2. Next, we performed conditional analyses of these clumped SNPs with the lead SNP for the interaction of T2D status with triglycerides. From this conditional analysis, we observed that none of the clumped SNPs abrogated the signal from the lead SNP.

 We have added a results section to the manuscript describing this additional analysis. 

Text in the manuscript:

To evaluate the independence of the lead variant, we clumped the 478 genome-wide significant variants and performed conditional analyses. After clumping, we retained 8 individual signals, and rs3957148 was in strong LD with the lead variant. However, conditional analyses with all these eight clumped variants for rs9274619:A interaction with T2D status did not abrogate the lead variant’s signal (Supplementary Table 2).

4. It would be interesting, though not required by journal to address the question. Is there a “… relationship between the HLA-DQB1*03:02 SNP and TG levels in the type 1 diabetes subjects in the UKBB “?

Author Response:

Thank you for this comment. Yes, there is a relationship between the lead SNP from this study and TG in T1D samples. In summary, the TG-lowering rs9274619:A was strongly associated with T1D after adjusting for T2D (p-value=8.6x10-113), but not significantly associated with T2D status after adjusting for T1D (p-value=0.29). However, when assessing for interactions on TGs, there was still a significant interaction with T2D independent of T1D (betainteraction=-0.055; pinteraction=5.28x10-9) and more strongly with T1D independently of T2D (betainteraction=-0.252; pinteraction=5.53x10-49). We included this in the manuscript text (lines 199-205) and tabulated the findings in the Supplementary Table 10.

5. Explain in the manuscript how other researchers can gain access to the data, note the PLOS one data policy.

Author Response:

Thank you for this comment. We have included a section describing the data availability for UK Biobank and MGB Biobank. 

Text in the manuscript:

Data availability:

Data cannot be shared publicly by the authors because of information governance restrictions around health data. The UK Biobank data can however be downloaded following a project approval process. Researchers wishing to access the data can apply directly to the UK Biobank https://www.ukbiobank.ac.uk/enable-your-research/apply-for-access and the process involves registering on the access management system, submitting a research study protocol and paying a fee directly to the UK Biobank. The Mass General Brigham Biobank (MGBB) individual-level data are available from https://personalizedmedicine.partners.org/Biobank/Default.aspx, where the data is available through institutional review board (IRB) approval, therefore not publicly available. Summary statistics from the current study will be shared through the Cardiovascular Disease Knowledge Portal (CVDKP) https://cvd.hugeamp.org/.

REVIEWERS COMMENTS:

Reviewer #1: Dear Author,

I strongly recommend this publication for the following perspectives. First, the large cohort sample sizes for both the cases and controls increases the statistical power for detection of variants with small effect sizes, a vital limitation in analysis of complex genetic traits, were the genetic effect is due to multiple variants with small effect sizes. Second, adjustment for BMI as a potential confounder is of high relevance giving the precision of the statistical model. Third, the implementation of Cochran statistical method in METAL, is highly recommended in conduction meta-analysis of GWAS.

Fourth, linking the GWAS results to the gene expression in different tissues associated with lipid metabolism strongly augment and support the finding. Fifth, even though identification of genetic association of T2D with the well-known T1D HLA locus adds to the complexity of the phenotype yet it could be of valuable information reflecting the protective role of immune system in this category of T2D patients. Overall, great research and work, well written, it adds value for understanding the heterogeneity of T2D supporting the fact that there are underlying subtypes and identifying the role of immune system in T2D not only T1D and diabetic complications.

Author Response:

We thank the reviewer for the positive comments.

Reviewer #2: Selvaraj et al. performed genome-wide association and interaction studies among UKB and MGBB subjects, they found that the HLA-DQB1 locus was associated with lower triglyceride levels in T2D patients, but not in non-T2D subjects.

Several issues need to be addressed before acceptance for publication: 

1) The affection status of T2D in UKB subjects was based on self-reported data, the incidence of T2D (5.0%) in UKB participants seems lower than general population, given the mean age of 56.6 years. Should fasting glucose and HbA1c be considered for the diagnosis of T2D? 

Author Response:

Thank you for this comment. UK Biobank is well recognized to have a healthy volunteer bias with generally lower prevalences of chronic conditions relative to the UK population at large (Fry A et al. Am J Epidemiol 2017). This contrasts with our healthcare associated biobank, MGB Biobank (prevalence 27.65%). Reassuringly, in both contexts, the findings hold. Consistent with prior studies, we have focused on clinician recognized and self reported T2D. To improve the T2D diagnosis sensitivity, we now consider HgA1c (i.e., >= 48 mmol/mol) for the case definition; unfortunately, fasting glucose is not available. Indeed in secondary analysis, we associated blood biomarkers including random glucose and HbA1c with the lead SNP and observe significant associations (Supplementary Tables 13 &14).

We investigated using HbA1c for the diagnosis of T2D status. As an additional analysis, we stratified the cohort based on HbA1c values (UKB data field: 30750), where HbA1c-based T2D had HbA1c >=48 mmol/mol. Using this criterion, we were able to identify 15,180 HbA1c-based T2D samples. The interaction model of the lead SNP with HbA1c-augmented T2D showed higher significance on TG (betainteraction= -0.159; pinteraction= 1.26x10-54). Through this classification, we were able to add 5212 samples more to the existing T2D stratified group based on self-reporting and ICD codes. We finally included the ~5K samples to the ~21K T2D group and tested the interaction modeled with the lead SNP to show a stronger significance (betainteraction= -0.102; pinteraction= 1.03x10-35). We included these results in the revised manuscript.

Text in the manuscript:

Since HbA1c is a strong predictor of T2D, HbA1c is the most significant biomarker interacting with the lead SNP (Supplementary Table 14). When also using HbA1c >= 48 mmol/mol to define T2D samples as described by Young et al (14), 15,180 samples were identified, with an additional 5212 samples to the previous T2D group. The interaction model of HbA1c-based only T2D with the lead SNP showed higher significance with TG (betainteraction= -0.159; pinteraction= 1.26x10-54). Next, we added the 5212 samples to our existing ICD10-based T2D groups and conducted a sensitivity analysis for the interaction. The addition of samples showed consistent significant interaction (betainteraction= -0.102; pinteraction= 1.03x10-35).

2) The "concomitant diagnosis of T1D among those with T2D" is confusing. It is highly unlikely that a subject had both T1D and T2D, however, it could be an issue since affection statuses were self-reported. Indeed, the HLA-DQB1 locus was only associated with T1D, not with T2D, in previous GWASs in large cohorts. Since no information of LADA diagnosis was available in UKB, testing LADA loci in UKB (the number of LADA patients was limited) provided little information. 

Author Response:

Thank you for this comment. We agree that overlapping samples between T1D and T2D are not well-defined and this also occurs clinically. In the current paper, we tried to dissect the association of this locus by comparing T1D, T2D, and presumed LADA (overlap of T1D and T2D). As shown in Supplementary Table 10, the lead variant has strongest association with T1D, followed by T2D. The number of samples with LADA is limited but the lead variant’s association is still genome-wide significant. We have discussed this in the manuscript lines 178-196 and added a sentence to emphasize the effect of overlapping samples.

Text in the manuscript:

We additionally checked the main effects of rs9274619:A for presumed LADA samples (N=2580) in our dataset and observed a significant association (p-value=8.37x10-41), showing that overlapping T1D and T2D samples could potentially contribute to the observed association of this locus.

3) It is interesting to know that the HLA-DQB1 polymorphisms were associated with higher HbA1c and glucose levels, higher risks of diabetic marco- and microvascular complications, but lower TG, urate, and body weight phenotypes in T2D subjects. From previous studies, seems TG was an independent risk factor of cardiovascular diseases in T2D patients, and a predictor of T2D itself. The authors may discuss the difference of the association in T2D and non-T2D subjects.

Author Response:

Thank you for this comment. We added a sentence in the discussion explaining the rationale for the stratified GWAS. Please check Editors comment #2 for the response. 

Reviewer #3: 

1. The study suffers from the habit of classifying the bulk of diabetes patients above 20-25 years of age as type 2 diabetes. The diagnosis is diabetes but the classification is type 2 diabetes based on loose clinical criteria. The strong association between low levels of TG and the A variant in the HLA-DQ B1*03:02 is a distinct example that the authors should discuss.

Author Response:

Thank you for this comment. We would like to point out that we defined T2D based on self-reported status and based on ICD10 codes. We recognize that clinicians may erroneously classify T2D for all adult-onset diabetes because it comprises the overwhelming majority. We have addressed a similar question, please check reviewer #2 comment #1 for the additional analysis that we conducted. In brief, we consider diagnostic overlaps for T1D and T2D and also consider undiagnosed T1D by a T1D PRS.

2. The authors should also discuss the lack of information on GADA or other islet autoantibodies, c-peptide levels as well as HbA1c and the weakness that most of the diabetes classification is self-reported.

Author Response:

Thank you for this comment. We agree that the classification of T2D status is complicated where multiple factors play a very important role and particularly the potential lack of LADA or T1D recognition. HgA1c is indeed available and we now include this in new analyses. We have added a sentence in the limitation section. Please also refer to the comments above about further sensitivity analyses now included.

Text in the manuscript:

Second, GADA or other islet autoantibodies and C-peptide levels are not present, despite using ICD codes and T1D polygenic risk score in sensitivity analyses, we cannot rule out the possibility of T1D-like features in some of the classified T2D cases. 

3. Genetic Risk Scores (GRS) for autoimmune type 1 diabetes has been developed using the UKBB (Oram and others) and these data should be run in parallel with the present analysis. It may resolve the many issues of re-classifying the patients with autoimmune type 1 diabetes rather than maintaining that they are type 2 diabetes patients.

Author Response:

Thank you for this comment and the valuable suggestion. We have integrated the PRS from the 67 SNPs reported by Oram et al to prioritize undiagnosed T1D samples. We show that the lead SNP in nominally significant in its interaction to T2D even after the removal of T1D samples based on ICD codes and T1D PRS. Please see our response in Editors comment #1.

4. The region around the reported DQB1*03:02 SNP should be further analysed to provide a heat-map of the region to indicate that the particular SNP reported has the highest risk, or p-value, compared to neighbouring SNPs. The authors need to provide further analysis that the present high risk SNP is not in linkage disequilibrium with other genetic factors.

Author Response:

Thank you for this comment and the valuable suggestion. We have addressed this comment with additional conditional analyses. Please see our response in Editors comment #3.

5. There are type 1 diabetes patients (usually more clearly defined when self-reported) in the UKBB. What is the risk of the present DQB1*03:02 SNP for type 1 diabetes? Is there data to indicate that this particular SNP marked low level TG also in type 1 diabetes patients?

Author Response:

Thank you for this comment. We have conducted this analysis. Please see our response in Editors comment #4.

---

## [Decision Letter · Decision Letter 1]

12 Sep 2022

PONE-D-22-11074R1Genome-wide Discovery for Diabetes-Dependent Triglycerides-Associated LociPLOS ONE

Dear Dr. Natarajan,

Thank you for submitting your manuscript to PLOS ONE. After careful consideration, we feel that it has merit but does not fully meet PLOS ONE’s publication criteria as it currently stands. Therefore, we invite you to submit a revised version of the manuscript that addresses the points raised during the review process.

few minor things need your attention, see reviewers comments.

We look forward to receiving your revised manuscript.

Kind regards,

Arnar Palsson, Ph.D.

Academic Editor

PLOS ONE

Journal Requirements:

Additional Editor Comments:

Thanks for responding so well to the reviewers suggestions, this is very nearly ready.

One reviewer suggest minor touch ups to the manuscript. Should be relatively quickly completed.

Reviewers' comments:

Reviewer's Responses to Questions

**Comments to the Author**

1. If the authors have adequately addressed your comments raised in a previous round of review and you feel that this manuscript is now acceptable for publication, you may indicate that here to bypass the “Comments to the Author” section, enter your conflict of interest statement in the “Confidential to Editor” section, and submit your "Accept" recommendation.

Reviewer #1: All comments have been addressed

Reviewer #3: (No Response)

2. Is the manuscript technically sound, and do the data support the conclusions?

Reviewer #1: Yes

Reviewer #3: Yes

3. Has the statistical analysis been performed appropriately and rigorously? 

Reviewer #1: Yes

Reviewer #3: Yes

4. Have the authors made all data underlying the findings in their manuscript fully available?

Reviewer #1: Yes

Reviewer #3: Yes

5. Is the manuscript presented in an intelligible fashion and written in standard English?

Reviewer #1: Yes

Reviewer #3: Yes

6. Review Comments to the Author

Reviewer #1: This study highlights the necessity of understanding the genetics of T2D and underlying pathways.

T2D is heterogeneous complex genetic trait, in order to proceed for personalized medical care for each patient, tailored medical care based on genetics, genetic risk scores offers valuable tool for patient T2D characterization and drug _repurposing.

Reviewer #3: The authors have answered my queries. It would strengthen the observed association to check if TG levels are related to the rs9274619 genotypes A/A, A/G and G/G. As rs9274619 is located between the HLA-DQB1 and HLA-DQA2 loci, the authors may want to refer this potentially quantitative trait loci as linked to HLA-DQB1/HLA-DQA2 rather than only HLA-DQB1.

LD to either gene may be about the same.

The authors should consider to add this recent investigation of small vessel vasculitis, an immune related disease also related to rs9274619: Dahlqvist et al. Identification and functional characterization of a novel susceptibility locus for small vessel vasculitis with MPO-ANCA. Rheumatology (Oxford). 2022 Aug 3;61(8):3461-3470.

Finally, HLA genotypes seemed to influence levels of TG in children at increased HLA genetic risk and positive for islet autoantibodies but prior to clinical diagnosis. TG levels were lower in the not-yet diabetes affected subjects and may indeed reflect the rs9274619 variant.

7. PLOS authors have the option to publish the peer review history of their article (what does this mean?). If published, this will include your full peer review and any attached files.

Reviewer #1: **Yes: **Dina Mansour Aly

Reviewer #3: No

---

## [Author Response · Author response to Decision Letter 1]

23 Sep 2022

Editors Comment:

Authors response:

Thank you for the comment. We have corrected one reference and removed one reference.

Reference no 14:

Young KG, McDonald TJ, Shields BM. Glycated haemoglobin measurements from UK Biobank are different to those in linked primary care records: implications for combining biochemistry data from research studies and routine clinical care. Int J Epidemiol. 2022 Jun 13;51(3):1022–4.

Reference no 27:

Removed

Reviewers' comments:

Reviewer #1: This study highlights the necessity of understanding the genetics of T2D and underlying pathways.

T2D is heterogeneous complex genetic trait, in order to proceed for personalized medical care for each patient, tailored medical care based on genetics, genetic risk scores offers valuable tool for patient T2D characterization and drug _repurposing.

Authors response:

Thank you for the positive comment.

Reviewer #3: The authors have answered my queries. It would strengthen the observed association to check if TG levels are related to the rs9274619 genotypes A/A, A/G and G/G. As rs9274619 is located between the HLA-DQB1 and HLA-DQA2 loci, the authors may want to refer this potentially quantitative trait loci as linked to HLA-DQB1/HLA-DQA2 rather than only HLA-DQB1.

LD to either gene may be about the same.

The authors should consider to add this recent investigation of small vessel vasculitis, an immune related disease also related to rs9274619: Dahlqvist et al. Identification and functional characterization of a novel susceptibility locus for small vessel vasculitis with MPO-ANCA. Rheumatology (Oxford). 2022 Aug 3;61(8):3461-3470.

Finally, HLA genotypes seemed to influence levels of TG in children at increased HLA genetic risk and positive for islet autoantibodies but prior to clinical diagnosis. TG levels were lower in the not-yet diabetes affected subjects and may indeed reflect the rs9274619 variant.

Authors response:

Thank you for the suggestion, we have added the below sentences to address the comments. Throughout the manuscript we have made changes to sentences referring the lead variants proximity to HLA-DQB1/DQA2. Additionally, we agree that our results showing the main effect of rs9274619 on TGs is consistent with the prior literature and here we show its dependency on diabetes status.

Line number 114: The mean raw TG measurements were significantly different between the reference and alternative genotypes in T2D samples (rs9274619(G/A or A/G) : meandiff=5.45 mg/dl; p-value=1.26x10-02; rs9274619(A/A) : meandiff=25.6 mg/dl; p-value=3.78x10-03). 

Line number 165: Since rs9274619:A located between both HLA-DQB1 and HLA-DQA2, the variant could be a potential quantitative trait loci to both the genes.

Line number 193: Interestingly, a recent investigation on Anti-neutrophil cytoplasmic antibody (ANCA)-associated vasculitides (AAV) by Dahlqvist et al have identified rs9274619 as a lead variant for myeloperoxidase (MPO)-ANCA association (11), showing its importance in immune-related vascular diseases.

Figures:

Authors response:

Thank you for the information. We have uploaded the figures through this tool.

---

## [Editor Report · Decision Letter 2]

26 Sep 2022

Genome-wide Discovery for Diabetes-Dependent Triglycerides-Associated Loci

PONE-D-22-11074R2

Dear Dr. Natarajan,

We’re pleased to inform you that your manuscript has been judged scientifically suitable for publication and will be formally accepted for publication once it meets all outstanding technical requirements.

Kind regards,

Arnar Palsson, Ph.D.

Academic Editor

PLOS ONE
---

## [Editor Report · Acceptance letter]

13 Oct 2022

PONE-D-22-11074R2 

Genome-wide Discovery for Diabetes-Dependent Triglycerides-Associated Loci 

Dear Dr. Natarajan:

I'm pleased to inform you that your manuscript has been deemed suitable for publication in PLOS ONE. Congratulations! Your manuscript is now with our production department. 

Kind regards, 

on behalf of

Dr. Arnar Palsson 

Academic Editor

PLOS ONE